# LightAVSeg: Lightweight Audio-Visual Segmentation

Qing Zhong[1]   Guodong Ding[2]*   Lingqiao Liu[3]   Zaiwen Feng[1]   Lin Yuanbo Wu[4][5]   Angela Yao[2]

## Abstract

Audio-Visual Segmentation (AVS) targets pixel level localization of sounding emitting objects in videos. However, existing models rely on dense cross-modal attention with quadratic computational cost, limiting their suitability for resource efficient deployment. Most efficiency oriented methods focus on backbone reduction and overlook the interaction module as the primary bottleneck. This paper proposes LightAVSeg, a lightweight framework that replaces heavy attention with a decoupled design for semantic filtering and spatial grounding, resulting in interaction costs that scale linearly with spatial resolution. Furthermore, we introduce an auxiliary alignment loss to enforce semantic consistency during training with zero inference overhead. Extensive experiments demonstrate that LightAVSeg achieves a new state-of-the-art among lightweight methods: with 20.5M parameters ($\sim 1/7$ of AVSegFormer), it reaches 50.4 mIoU on the MS3 benchmark and enables efficient inference on a mobile processor.

## 1. Introduction

Audio-Visual Segmentation (AVS) (Zhou et al., 2022) aims to delineate sound-emitting objects in video sequences at the pixel level. Compared to conventional sound source localization that produces coarse heatmaps (Arandjelovic & Zisserman, 2017; 2018; Chen et al., 2021; Cheng et al., 2020; Owens & Efros, 2018; Senocak et al., 2018), AVS requires accurate object boundaries, which demands effective integration of acoustic cues with visual semantics. State-of-the-art (SOTA) approaches (Gao et al., 2024; Li et al., 2024) excel by modeling cross-modal dependencies via

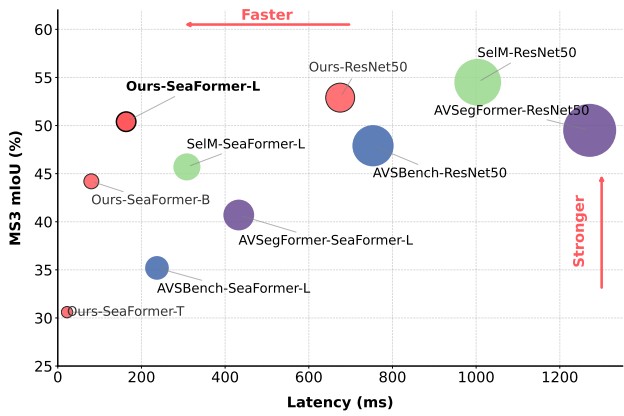

*Figure 1.* Efficiency-Accuracy Trade-off. We evaluate all models on the MS3 benchmark, and inference speeds are measured on a Snapdragon 8 Elite mobile CPU. Our method LightAVSeg outperforms heavy SOTA methods in both speed and accuracy on mobile device.

Transformer-based fusion. However, these gains often come at the expense of efficiency. SOTA systems built upon heavy visual backbones like ResNet-50 (He et al., 2016) or Pyramid Vision Transformer (PVT) variants (Wang et al., 2021; 2022) and attention-based interaction modules result in considerable computational cost. For instance, processing a video of $224 \times 224$ resolution with AVSegFormer (Gao et al., 2024) takes more than 1.2 seconds per frame on a high-end mobile processor such as Snapdragon 8 Elite, as illustrated in Fig. 1. Such requirements have made deployment challenging on resource-constrained platforms such as mobile devices and VR headsets, limiting the potential for efficient applications like on-device video editing and augmented reality.

Lightweight AVS aims to enable efficient inference on mobile devices. However, simply replacing backbones with mobile-friendly alternatives (*e.g.*, MobileNetV2 (Kong et al., 2020)) does not fully resolve the efficiency issue, because the cross-modal interaction module can still dominate latency and memory. Standard attention-style fusion requires dense token interactions, where cost grows quadratically with the number of tokens. Since $N$ scales with the feature resolution ($N \propto H \times W$), the complexity of $\mathcal{O}(N^2)$ becomes especially costly at higher resolutions. This motivates a lightweight interaction design that avoids dense

[1]College of Informatics, Huazhong Agricultural University, Wuhan, China [2]School of Computing, National University of Singapore, Singapore [3]School of Computer Science, Adelaide University, Australia [4]School of Engineering, University of Warwick, Coventry, UK [5]Zhejiang Yuexiu University, Shaoxing, China. Correspondence to: Guodong Ding <dinggd @comp.nus.edu.sg>.

*Proceedings of the 43rd International Conference on Machine Learning*, Seoul, South Korea. PMLR 306, 2026. Copyright 2026 by the author(s).

pairwise token computations while preserving effective audio guidance.

A key observation in AVS is that fine-grained localization boundaries are largely encoded in visual spatial features, while audio primarily provides global semantic cues to identify which object is sounding. This implies that computing dense pixel-to-pixel affinity matrices can be redundant, as global auditory semantics are sufficient to select relevant visual channels. Motivated by this, we propose LightAVSeg, a compact framework with a Reciprocal Audio-Visual Encoder and a Cross-Modal Fusion Decoder to decouple the interaction. The Reciprocal Audio-Visual Encoder focuses on semantic filtering (what) by iteratively refining a global audio state, whereas the Cross-Modal Fusion Decoder handles spatial grounding (where) by injecting these cues into the visual hierarchy. Crucially, this design replaces dense attention ($\mathcal{O}(N^2)$) with lightweight channel-wise modulations, achieving strict linear complexity ($\mathcal{O}(N)$).

Furthermore, lightweight models constrained by reduced capacity are susceptible to learning spurious cross-modal correlations (Wan et al., 2023; Zhang et al., 2022; 2024). To mitigate this, we introduce an auxiliary alignment loss, which enforces an explicit, pixel-wise alignment between the global audio cue and valid visual regions. The alignment acts as a strong structural prior, explicitly guiding the decoder to focus on the sounding object during training. Crucially, this knowledge is distilled into the network weights, allowing the auxiliary branch to be discarded during inference with zero extra cost.

Our evaluation on three common AVS benchmarks highlights that LightAVSeg achieves the best accuracy-efficiency balance (see Fig. 1). With a compact model size of 20.5M parameters ($\sim 1/7$ of AVSegFormer-R50), it reaches 50.4 mIoU on MS3, comparable to heavier ResNet-50 based counterparts. On a Snapdragon 8 Elite mobile CPU, our model also achieves a latency of 163.4 ms, about $8\times$ speedup over the AVSegFormer (Gao et al., 2024). This provides practical latency for interactive on-device editing and helps bridge the gap between research models and practical mobile deployment.

Our main contributions are summarized as follows: 1) We propose LightAVSeg, a mobile-friendly framework that decouples interaction into a Reciprocal Audio-Visual Encoder for semantic filtering and a Cross-Modal Fusion Decoder for spatial grounding. This design replaces quadratic attention with hierarchical channel-wise modulation, achieving linear complexity ($\mathcal{O}(N)$) while preserving effective guidance. 2) We introduce an auxiliary Multi-Scale Audio-Visual Alignment Loss ($\mathcal{L}_{msa}$) to mitigate semantic ambiguity in lightweight networks and guide the decoder to determine where to segment. 3) Our framework achieves state-of-the-art lightweight Audio-Visual Segmentation performance on three AVS benchmarks and enables efficient inference on mobile devices.

## 2. Related Work

### 2.1. Semantic Segmentation

Semantic segmentation has evolved from convolution-based models such as FCN (Long et al., 2015), DeepLab (Chen et al., 2017), and edge-aware medical segmentation networks (Zhou et al., 2025b), which focus on local context and boundary details via dilated convolution (Wu et al., 2022) and edge-aware feature aggregation, to Transformer-based frameworks such as SegFormer (Xie et al., 2021) and CTVIS (Ying et al., 2023) that model global dependencies in image and video segmentation. Beyond architectural advances, contrastive learning methods such as PiCo (Zhou & Wang, 2024) enhance representation by enforcing pixel-wise discrimination across images, while prototype-based segmentation (Xie et al., 2021) interprets classifiers as learnable prototypes for better generalization. Recent works including LLMFormer (Shi et al., 2025) and knowledge-guided SAM-based methods (Wu & Liu, 2026) further extend segmentation to the open-vocabulary setting by leveraging language priors and foundation segmentation models. In video understanding, online temporal segmentation and video instance segmentation pre-training have also emphasized temporal dependency modeling and temporally consistent representations (Zhong et al., 2024; 2025). Meanwhile, related structured visual generation and animation studies explore rendering-aware feedback, sparse state modeling, and motion priors for vector graphics and sketch animation (Liang et al., 2026a;b; 2025), providing complementary perspectives on structure-preserving visual representation learning.

### 2.2. Audio-Visual Segmentation

Audio-visual segmentation (AVS) (Zhou et al., 2022; 2025a) extends sound source localization to pixel-level understanding. While early CNN-based models (Zhou et al., 2022) suffered from limited receptive fields, Transformer-based frameworks like AVSegFormer (Gao et al., 2024) established a new paradigm by leveraging cross-modal attention for global feature fusion. Subsequent works have further improved robustness (Li et al., 2024; Gong et al., 2025b) or sequence modeling efficiency (Gong et al., 2025a). However, these methods often overlook the high computational cost of the interaction module itself. Current state-of-the-art approaches still rely on dense attention mechanisms with quadratic complexity, creating a significant bottleneck for deployment on resource-constrained edge devices.

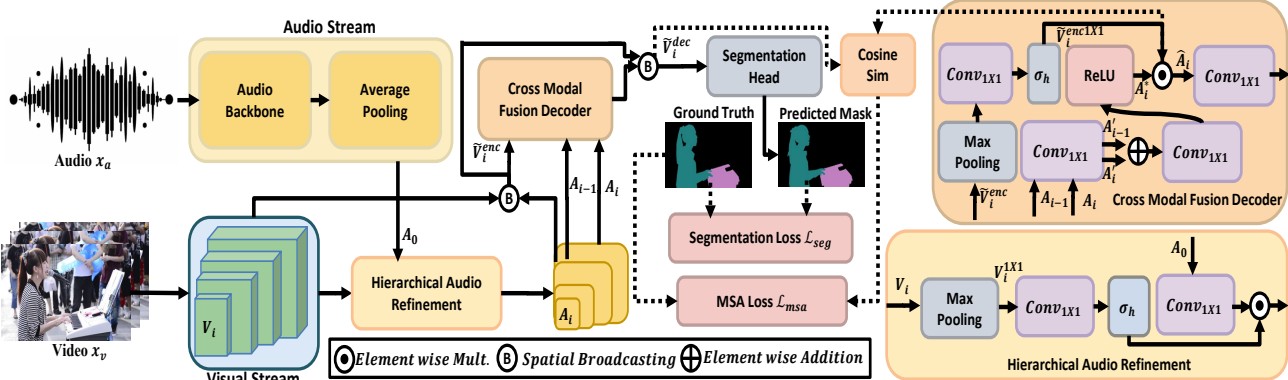

*Figure 2.* Overview of LightAVSeg. Following the visual and audio streams, we introduce the Reciprocal Audio-Visual Encoder to iteratively refine the global audio state using visual context, the Cross-Modal Fusion Decoder to inject these auditory cues back into the visual stream for segmentation, and the Multi-Scale Audio-Visual Alignment Loss ($\mathcal{L}_{msa}$) to enforce progressive cross-modal consistency.

## 2.3. Mobile Vision Transformers

Mobile semantic segmentation focuses on achieving accurate dense prediction under strict efficiency and memory constraints on edge devices. Early lightweight CNNs such as BiSeNet (Nie et al., 2023) and Fast-SCNN (Poudel et al., 2019) introduced dual-branch and shared-computation designs to balance accuracy and latency. With the advent of mobile vision transformers, TopFormer (Zhang et al., 2022) combined token pyramid structures and convolutional stems to capture global dependencies efficiently, while SeaFormer (Wan et al., 2023) proposed a squeeze-enhanced axial attention that reduces the quadratic complexity of self-attention for high-resolution inputs. Extending this paradigm beyond image segmentation, MobileInst (Zhang et al., 2024) applies lightweight dual-transformer decoders for mobile video instance segmentation, demonstrating that compact transformer architectures can effectively generalize to spatiotemporal dense prediction on mobile CPUs.

## 3. Method

### 3.1. Overall Architecture

LightAVSeg adopts a compact dual-stream architecture tailored for efficient audio-visual segmentation. Fig. 2 shows the framework; there is a visual and an audio stream, both lightweight, and a hierarchical fusion pathway that progressively integrates audio cues into visual representations.

Given an input video $x_v$, the visual stream extracts hierarchical feature maps $\{V_i\}_{i=1}^N$ using a convolution transformer encoder. In parallel, the audio stream transforms the synchronized waveform $x_a$ into a log-mel spectrogram, which is then processed by an audio backbone. The encoder aggregates spectral features within each frame-synchronized window, producing a spatially global but temporally aligned audio state $A_0$. This design preserves rich spatial structure

in the visual branch while keeping the audio representation synchronized.

For efficient interaction, we design a Reciprocal Audio-Visual Encoder that propagates the global audio state across network stages. Instead of spatial attention, this module performs hierarchical bidirectional updates: it refines the global audio semantic tokens using visual context and subsequently modulates the visual features via channel-wise injection. This design ensures that global sound cues guide visual feature selection without dense spatial affinity matrices.

Following the encoder, a Cross-Modal Fusion Decoder is used to generate the final segmentation mask. It progressively upsamples the audio enhanced visual features, while continuously injecting the refined audio state to maintain semantic consistency during resolution recovery.

The entire network is trained end-to-end with a compound objective. We use a primary segmentation loss $\mathcal{L}_{\text{seg}}$ combining Dice (Gao et al., 2024) and BCE (Zhou et al., 2022) terms to supervise the pixel-wise prediction:

$$\mathcal{L}_{\text{seg}} = \mathcal{L}_{\text{dice}} + \mathcal{L}_{\text{bce}}. \quad (1)$$

In addition to $\mathcal{L}_{\text{seg}}$, we further introduce an auxiliary Multi Scale Audio-Visual Alignment Loss to enforce cross-modal consistency, the details of which are elaborated in Sec. 3.5.

Our framework establishes a clear functional division: the Reciprocal Audio-Visual Encoder focuses on identifying the sounding object via global semantic filtering, while the Cross-Modal Fusion Decoder determines the segments by grounding these cues into pixel-level boundaries.

### 3.2. Multi-modal Representation

Given a synchronized audio-visual sequence comprising visual frames $x_{\text{v}} \in \mathbb{R}^{T \times 3 \times H \times W}$ and a raw audio wave-

form $x_a \in \mathbb{R}^{N_a}$, where $T$ denotes the number of frames and $N_a$ represents the sampled audio duration, our goal is to learn efficient, modality-consistent representations for audio-visual segmentation. Both streams are processed by mobile-friendly backbones that capture complementary semantics from spatial and acoustic cues.

**Visual Stream.** The visual encoder follows a hybrid convolution-transformer design that progressively extracts hierarchical representations. Specifically, the input frames $x_v$ are processed by a series of blocks to produce a set of scale-aware feature maps, forming a feature pyramid $\{V_i\}_{i=1}^N$. Each $V_i \in \mathbb{R}^{B \times C_i \times H_i \times W_i}$ captures semantics at a specific scale (e.g., strides of $2^{i+1}$). The early stages utilize convolutions for local details, while the later stages use transformer blocks to encode long-range dependencies. These hierarchical embeddings $\{V_i\}$ provide the necessary multi-scale spatial foundation for the subsequent layer-wise audio-visual interaction.

**Audio Stream.** The audio encoder adopts a convolutional network based on MobileNetV2 to generate spectral embeddings. First, the resampled 16 kHz mono audio waveform $x_a$ is processed via a Short-Time Fourier Transform to generate a log-mel spectrogram $x_{mel} \in \mathbb{R}^{N_a \times 96 \times 64}$ with $F{=}64$ mel bins. This spectrogram is fed into the audio backbone (e.g., MobileNetV2 (Kong et al., 2020)), which aggregates frequency and local temporal context to yield audio features $A \in \mathbb{R}^{T \times C_a}$. To initialize the hierarchical interaction, we reshape these embeddings to obtain the initial audio state $A_0 \in \mathbb{R}^{T \times C_a \times 1 \times 1}$. Note that our $A_0$ maintains one-to-one temporal correspondence with the visual frames ($T$ steps), to serve as a synchronized global prior for subsequent spatial interaction.

### 3.3. Reciprocal Audio-Visual Encoder

Standard AVS methods (Zhou et al., 2022; Gao et al., 2024) often delay modality fusion until the decoding stage. However, SelM (Li et al., 2024) revealed that early interaction is crucial for suppressing visually irrelevant sound associations (e.g., background noise) before they propagate. Therefore, instead of static audio features, we propagate a dynamic audio state that evolves layer-by-layer under visual guidance.

**Hierarchical Audio Refinement.** Given the audio state from the previous layer $A_{i-1}$ and the current visual feature $V_i$, we extract a global visual descriptor from $V_i$ to guide the audio update. Concretely, we obtain $V_i^{1x1}$ by spatially max pooling $V_i$ to $1{\times}1$. Collapsing the spatial dimensions forces the interaction to discard location information and limits the encoder's role to semantic selection rather than spatial localization. This operation is structurally efficient: unlike standard cross-modal attention layers (Gao et al., 2024) that

compute dense affinity matrices with quadratic complexity $\mathcal{O}(N^2)$ (where $N{=}H_i \times W_i$), our global interaction avoids any token-to-token affinity. Instead, it relies solely on pointwise projections and broadcasting, leading to strict linear scaling w.r.t. spatial tokens ($\mathcal{O}(N)$). This minimizes the parameter overhead while preserving the semantic selectivity needed to filter audio features. Both $A_{i-1}$ and $V_i^{1x1}$ are mapped via $1{\times}1$ convolutions and fused through gated reweighting:

$$A_i = \text{Conv}_{1\times1}(A_{i-1}) \odot \sigma_h\big(\text{Conv}_{1\times1}(V_i^{1x1})\big), \quad (2)$$

Here, $\sigma_h(\cdot)$ is the h-sigmoid activation, and $\odot$ indicates element-wise multiplication. The visual descriptor $V_i^{1x1}$ acts as a semantic gate: it suppresses audio channels that correspond to objects not visible in the current frame, ensuring $A_i$ becomes increasingly scene-specific.

By repeatedly refining the global audio token under visual guidance, the model gradually makes the audio state more scene-aware while avoiding the construction of spatial audio maps. This hierarchical update allows sound cues to evolve smoothly with visual depth, offering a globally consistent representation that adapts to changing visual semantics without increasing spatial cost.

**Audio-Guided Visual Enhancement.** The refined audio state $A_i$ encodes global semantic information but does not contain spatial structure. To inject such information into the visual stream without introducing spatial noise, we adopt a channel-wise modulation strategy.

Concretely, $A_i$ is already a global audio state with the stage-matched channel dimension. We simply broadcast it to the spatial resolution of $V_i$ and fuse it through residual addition:

$$\widetilde{V}_i^{enc} = V_i + \mathcal{B}(A_i), \quad (3)$$

where $\mathcal{B}(\cdot)$ denotes spatial broadcasting to $H_i \times W_i$. This treats audio as a global channel-wise bias while leaving spatial localization to the visual stream. Unlike spatial audio attention that requires constructing dense audio-conditioned maps and heavy projection layers, our fusion injects only a global bias via a lightweight broadcast operation. This design is parameter-free in terms of spatial transformation, ensuring that the visual feature enhancement incurs negligible FLOPs. In practice, it helps when the sounding-related channels are visually weak (e.g., small or occluded), since the audio signal consistently boosts the relevant channels before spatial decoding.

**Progressive Representation Flow.** The refinement proceeds in a hierarchical manner, where the fused audio feature $A_i$ serves as the prior for the next stage $A_{i+1}$. This propagation enables hierarchical reasoning, allowing the model to progressively refine the cross-modal representation from global semantics to fine-grained spatial cues

across stages. The final outputs $\{\widetilde{V}_i^{enc}, A_i\}$ form a hierarchical audio-visual encoder, providing strong contextual and acoustic foundations for subsequent fusion and segmentation. Crucially, since the visual input for the next stage $V_{i+1}$ is computed from the audio enhanced feature $\widetilde{V}_i^{enc}$, our visual encoder effectively "listens" to the audio throughout the entire feature extraction process, ensuring early suppression of visually irrelevant regions.

### 3.4. Cross-Modal Fusion Decoder

Audio-visual segmentation requires associating sound sources with their spatial extent. However, raw audio features often lack spatial grounding, while visual features alone are ambiguous when sounding objects are visually occluded or small. To address this mismatch, our decoder progressively injects audio cues into hierarchical visual features while maintaining temporal consistency across layers.

**Audio-Visual Fusion.** During the upsampling process, visual features undergo significant scale changes, which risks diluting the global semantic consistency established in the encoder. To this end, we maintain a recurrent audio state path parallel to the visual upsampling process. At stage $i$, the previous decoder audio state $A_{i-1}$ and the current encoder audio feature $A_i$ are first projected to a shared embedding space and concatenated. A lightweight gating operation then produces an updated audio representation:

$$A_i^* = \text{ReLU}\big(\text{Conv}_{1\times 1}\big([A'_{i-1}, A'_i]\big)\big), \qquad (4)$$

where $[\cdot, \cdot]$ denotes channel-wise concatenation, $A'_{i-1}$ and $A'_i$ denote channel-aligned audio embeddings.

To incorporate visual guidance, we compress the corresponding visual feature $\widetilde{V}_i^{enc}$ into a global descriptor and use it to reweight the fused audio state in a channel-wise manner:

$$\hat{A}_i = A_i^* \odot \sigma_h\big(\text{Conv}_{1\times 1}(\widetilde{V}_i^{enc1\times 1})\big), \qquad (5)$$

where $\widetilde{V}_i^{enc1\times 1}$ is a spatially max pooled $\widetilde{V}_i^{enc}$ to $1\times 1$. The updated $\hat{A}_i$ is then propagated to the next decoder stage.

We inject $\hat{A}_i$ into the visual pathway through a lightweight broadcast residual, treating audio as a global channel bias without introducing spatial noise:

$$\widetilde{V}_i^{dec} = \widetilde{V}_i^{enc} + \mathcal{B}\big(\text{Conv}_{1\times 1}(\hat{A}_i)\big), \qquad (6)$$

where $\mathcal{B}(\cdot)$ broadcasts a $1\times 1$ tensor to match the spatial size of $\widetilde{V}_i^{dec}$. This fusion allows temporally consistent sound cues to modulate visual feature selection while leaving spatial localization to the visual stream.

By maintaining a recurrent audio state and repeatedly grounding it with visual context, this module stabilizes the

audio-guided localization against resolution changes. Such design is well suited for audio-visual segmentation, where sound cues are global and temporally coherent, while visual evidence varies significantly across scales.

### 3.5. Loss Functions

Besides supervising the final predicted mask, we introduce an auxiliary Multi-Scale Audio-Visual Alignment Loss ($\mathcal{L}_{\text{msa}}$) that directly links audio cues to spatial visual activations at multiple scales. Specifically, given the decoder's visual features $\widetilde{V}_i^{dec}$ at scale $i$ and the paired global audio state $\hat{A}_i$, we first perform $\ell_2$ normalization:

$$\bar{a}_i = \frac{\hat{A}_i}{\|\hat{A}_i\|_2 + \epsilon}, \quad \bar{v}_i = \frac{\widetilde{V}_i^{dec}}{\|\widetilde{V}_i^{dec}\|_2 + \epsilon}, \qquad (7)$$

where $\bar{v}_i$ is normalized along the channel dimension for each spatial location, and $\bar{a}_i$ is the corresponding normalized global audio embedding.

We then compute a spatial cosine similarity map $\text{sim}_i = \langle \bar{v}_i, \bar{a}_i \rangle \in [-1, 1]$. This map is sharpened by a temperature $\tau$ and mapped to a probability-like score $s_i \in [0, 1]$:

$$s_i = \text{sigmoid}\left(\frac{\text{sim}_i}{\tau}\right). \qquad (8)$$

To apply supervision at a unified resolution, we bilinearly upsample $s_i$ to the ground truth mask size $(H, W)$, denoted as $\hat{s}_i \in \mathbb{R}^{B\times 1\times H\times W}$. For datasets providing multi-class masks $Y$, we form a binary sounding foreground mask $M = \mathbb{1}(\sum_{k=1}^K Y_k > 0)$; otherwise $M = Y$ for binary annotations. Finally, we supervise the audio-visual alignment using a pixel-wise Binary Cross Entropy (BCE) loss averaged across scales:

$$\mathcal{L}_{\text{msa}} = \frac{1}{S} \sum_{i=1}^S \text{BCE}(\hat{s}_i, M), \qquad (9)$$

where $S$ denotes the number of feature scales (*e.g.*, the last 3 stages), and $\hat{s}_i$ is the upsampled similarity map at scale $i$.

**Mechanism of Progressive Refinement.** Inspired by deep supervision strategies (Lee et al., 2015; Xie & Tu, 2015), our multi-scale objective forces early layers to establish global semantic correspondence to suppress noise, while compelling deeper stages to focus on fine-grained boundary localization. This mechanism drives the *global-to-local* evolution observed in Fig. 3, ensuring that the audio state accurately modulates visual features with increasing precision throughout the network depth.

We explicitly opt for a BCE-based alignment objective over the popular Kullback-Leibler (KL) divergence used in AVS-Bench (Zhou et al., 2022), as recent studies in distribution alignment (Shah et al., 2025) highlight that KL estimation

*Table 1.* Comparison with state-of-the-art methods on the S4 and MS3 settings.

| Method | Backbone | | GPU | Mobile | S4 | | MS3 | | Params |
|---|---|---|---|---|---|---|---|---|---|
| | Visual | Audio | ms ↓ | ms ↓ | $M_J$ ↑ | $M_F$ ↑ | $M_J$ ↑ | $M_F$ ↑ | |
| AVSBench (Zhou et al., 2022) | R50 | VGGish | 21.2 | 753.5 | 72.8 | 84.8 | 47.9 | 57.8 | 91.4M |
| AVSegFormer (Gao et al., 2024) | R50 | VGGish | 29.0 | 1271.4 | 76.5 | 85.9 | 49.5 | 62.8 | 151.1M |
| SelM (Li et al., 2024) | R50 | VGGish | 25.3 | 1003.8 | 76.6 | 86.2 | 54.5 | 65.6 | 117.6M |
| AVSBench (Zhou et al., 2022) | Sea | MNetV2 | 19.5 | 237.1 | 47.9 | 64.5 | 35.2 | 44.1 | 30.2M |
| AVSegFormer (Gao et al., 2024) | Sea | MNetV2 | 22.2 | 432.6 | 53.8 | 71.4 | 40.7 | 50.7 | 51.0M |
| SelM (Li et al., 2024) | Sea | MNetV2 | 18.7 | 308.6 | 59.1 | 77.4 | 45.7 | 57.6 | 39.5M |
| **Ours** | Sea | MNetV2 | **15.9** | **163.4** | **75.6** | **86.2** | **50.4** | **62.6** | **20.5M** |

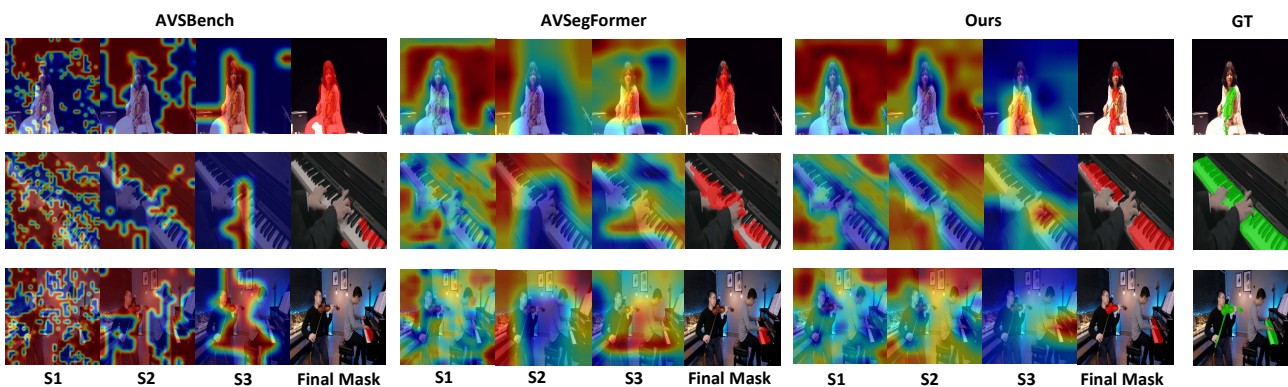

*Figure 3.* Visual comparison of feature activation maps on the MS3 benchmark. For each method, we visualize the features from the last three stages $S$ (from left to right). AVSBench is consistently distracted by background noise across stages. AVSegFormer focuses on the target but lacks boundary definition. Ours demonstrates a coarse-to-fine evolution, progressively suppressing background context to achieve precise alignment with the GT.

can be of high variance and sensitive to normalization strategies. Since our goal is to strictly align the high-response regions of the similarity map with the foreground mask, a pixel-wise BCE on the bounded scores provides a more stable supervision signal. The total loss is formulated as:

$$\mathcal{L} = \mathcal{L}_{\text{seg}} + \lambda \mathcal{L}_{\text{msa}}, \qquad (10)$$

where $\mathcal{L}_{\text{seg}}$ is the primary segmentation loss and $\lambda$ is a balance weight.

## 4. Experiments

### 4.1. Datasets and Metrics

**Datasets.** We evaluate our method on the AVS dataset (Zhou et al., 2022; 2025a), which serves as the standard benchmark for pixel-level audio-visual segmentation. It contains video clips with pixel-wise annotations sampled at one-second intervals. AVSBench is organized into three subsets targeting different levels of complexity, the Single-Source (S4) subset focuses on single-subject scenarios and contains 4,932 videos (5s duration) covering 23 categories. Following standard protocols, it is split into 3,452, 740, and

740 videos for training, validation, and testing, respectively. The Multi-Source (MS3) designed for complex auditory scenes with concurrent sound sources, and includes 424 videos. The split comprises 296 training, 64 validation, and 64 testing videos. Both S4 and MS3 provide binary masks for sounding object localization. Audio-Visual Semantic Segmentation (AVSS) To assess semantic understanding, we also utilize the AVSBench-Semantic subset. This large-scale subset extends the video duration to 10 seconds and scales up to 12,356 videos spanning 70 categories. Unlike S4/MS3, AVSS requires the model to predict pixel-wise semantic class labels for sounding objects.

**Evaluation Metrics.** We quantify performance using the Mean Jaccard Index and F-score. Mean Jaccard Index ($\mathcal{M}_{\mathcal{J}}$) (Everingham et al., 2010) commonly referred to as mIoU, measures the region-based segmentation accuracy by calculating the intersection-over-union between the predicted mask and the ground truth. Mean F-score ($\mathcal{M}_{\mathcal{F}}$) evaluates the contour precision of the segmentation results. In addition, to assess model efficiency, we report the number of parameters and inference latency on mobile and GPU.

*Table 2.* Comparison on the AVSS benchmark.

| Method | Backbone | | AVSS | |
| --- | --- | --- | --- | --- |
| | Visual | Audio | $M_J \uparrow$ | $M_F \uparrow$ |
| AVSBench | R50 | VGGish | 20.2 | 25.2 |
| AVSegFormer | R50 | VGGish | 24.9 | 29.3 |
| SelM | R50 | VGGish | 31.9 | 37.2 |
| AVSBench | Sea | MNetV2 | 10.0 | 11.7 |
| AVSegFormer | Sea | MNetV2 | 15.8 | 18.2 |
| SelM | Sea | MNetV2 | 20.6 | 26.6 |
| **Ours** | Sea | MNetV2 | **30.6** | **36.4** |

### 4.2. Implementation Details

We implement LightAVSeg in PyTorch using a pre-trained SeaFormer-Large (Wan et al., 2023) visual backbone and a frozen MobileNetV2 (Kong et al., 2020) audio backbone. Input images are resized to $224 \times 224$. Training runs for 60 epochs on a single RTX 3090 GPU using the AdamW optimizer (batch size 8, learning rate $1 \times 10^{-4}$), with hyperparameters set to $\tau = 0.1$ and $\lambda = 0.5$; other settings follow (Gao et al., 2024). To evaluate edge efficiency, we measure inference latency on a Snapdragon 8 Elite CPU using the TNN framework[1].

### 4.3. Comparison with State-of-the-Art Methods

**Performance on S4 and MS3 Benchmarks.** Table 1 compares LightAVSeg with state-of-the-art. On the S4 benchmark, our model establishes a new lightweight SOTA with 75.6 mIoU, outperforming the baseline (AVSegFormer-Sea) by +21.8 mIoU. Notably, it rivals the heavy AVSegFormer-R50 (76.5 mIoU) with only $\sim 1/7$ of the parameters (20.5M vs. 151.1M). On the complex MS3 benchmark, LightAVSeg achieves 50.4 mIoU, surpassing even the heavy AVSegFormer-R50 (49.5 mIoU). This suggests that our global interaction is more effective than dense cross-attention in suppressing noise and ambiguity. Regarding efficiency, LightAVSeg runs at 15.9 ms on GPU and 163.4 ms on a Snapdragon 8 Elite. This mobile latency is $7.8\times$ faster than AVSegFormer-R50 and $2.6\times$ faster than the naive baseline, confirming that linear-complexity interaction—rather than simple backbone replacement—is essential for efficient mobile deployment.

**Performance on AVSS Benchmark.** As shown in Table 2, we further evaluate semantic understanding capabilities on the large-scale AVSS dataset (Zhou et al., 2025a). Lightweight models typically struggle here due to limited capacity. However, LightAVSeg achieves 30.6 mIoU, doubling the performance of the AVSegFormer-Sea baseline

*Table 3.* Comparison of visual backbones on the MS3 benchmark.

| Backbone | latency | FLOPs | MS3 | |
| --- | --- | --- | --- | --- |
| | ms $\downarrow$ | G $\downarrow$ | $M_J \uparrow$ | $M_F \uparrow$ |
| ResNet-50 | 675.0 | 81.27 | 52.9 | 64.2 |
| SeaFormer-Tiny | 22.1 | 0.30 | 30.6 | 42.8 |
| SeaFormer-Base | 80.2 | 1.01 | 44.2 | 56.8 |
| SeaFormer-Large | 163.4 | 2.70 | 50.4 | 62.6 |

*Table 4.* Comparison of audio backbones on the MS3 benchmark.

| Backbone | latency | FLOPs | MS3 | |
| --- | --- | --- | --- | --- |
| | ms $\downarrow$ | G $\downarrow$ | $M_J \uparrow$ | $M_F \uparrow$ |
| VGGish | 659.1 | 4.37 | 51.5 | 63.8 |
| YamNet | 150.4 | 2.77 | 47.8 | 60.7 |
| MobileNetV2 | 163.4 | 2.70 | 50.4 | 62.6 |

(15.8 mIoU) and approaching the performance of the heavy SelM-R50 (31.9 mIoU). This demonstrates that our Reciprocal Audio-Visual Encoder and Alignment Loss effectively capture semantic concepts despite the compact model size.

### 4.4. Ablation Studies

**Impact of Visual Backbones.** In Table 3, we analyze the trade-off across different visual encoders. For a fair comparison, the ResNet-50 baseline is implemented using the AVSegFormer (Gao et al., 2024) architecture with a MobileNetV2 (Kong et al., 2020) audio backbone. While ResNet-50 (He et al., 2016) yields the highest accuracy (52.9 mIoU), its mobile latency is prohibitive (675.0 ms). On the other end of the spectrum, the SeaFormer-Tiny (Wan et al., 2023) achieves exceptional real-time speed (22.1 ms) but fails to capture sufficient semantic detail for the complex AVS task, resulting in a significant performance drop to 30.6 mIoU. Moving up to SeaFormer-Base improves accuracy to 44.2 mIoU (80.2 ms), yet a gap remains. Our selected backbone, SeaFormer-Large, strikes the optimal balance. It recovers the majority of the performance (50.4 mIoU, comparable to ResNet-50) while maintaining a feasible mobile latency (163.4 ms), validating its suitability for high-quality on-device segmentation.

**Impact of Audio Backbones.** Table 4 investigates the influence of the audio encoder. The standard VGGish (Li et al., 2020) backbone used in prior works is computationally heavy, with a latency of 659.1 ms. YamNet[2] offers low latency (150.4 ms) but degrades performance significantly (47.8 mIoU). MobileNetV2 (Kong et al., 2020) provides a comparable speed to YamNet (163.4 ms) while maintaining

---

[1]TNN: a uniform deep learning inference framework.

[2]YamNet: a lightweight audio event classifier.

*Table 5.* Comparison of loss functions on the MS3 benchmark.

| Loss function | $M_J \uparrow$ | $M_F \uparrow$ |
|---|---|---|
| $\mathcal{L}_{\text{seg}}$ | 49.3 | 60.8 |
| $\mathcal{L}_{\text{seg}} + \mathcal{L}_{\text{AVM}}$ | 49.2 | 61.2 |
| $\mathcal{L}_{\text{seg}} + \mathcal{L}_{\text{mix}}$ | 48.8 | 60.7 |
| $\mathcal{L}_{\text{seg}} + \mathcal{L}_{\text{msa}}$ | 50.4 | 62.6 |

*Table 6.* Ablation study of different fusion strategies on the MS3 benchmark.

| AGVE | CMFD | HAR | $\mathcal{L}_{\text{msa}}$ | $M_J \uparrow$ | $M_F \uparrow$ |
|---|---|---|---|---|---|
| | | | | 44.4 | 56.8 |
| ✓ | | | | 46.8 | 58.1 |
| ✓ | ✓ | | | 48.3 | 59.1 |
| ✓ | | ✓ | | 49.3 | 61.2 |
| ✓ | ✓ | ✓ | ✓ | 50.4 | 62.6 |

*Table 7.* Comparison of lightweight interaction designs on the MS3 benchmark.

| Interaction design | $M_J \uparrow$ | $M_F \uparrow$ |
|---|---|---|
| Strided attention | 30.6 | 39.3 |
| Top-$k$ attention | 44.7 | 57.2 |
| Ours | **50.4** | **62.6** |

*Table 8.* MS3 performance under different source-complexity levels. Each cell reports $M_J/M_F$.

| Method | 1 source | 2 sources | $> 3$ sources |
|---|---|---|---|
| AVSegFormer | 44 / 53 | 34 / 45 | 32 / 49 |
| Ours | **53 / 64** | **45 / 58** | **44 / 65** |

high accuracy (50.4 mIoU), justifying its selection as our audio stream encoder. Note that for all experiments, we initialize the backbone with AudioSet (Gemmeke et al., 2017) pre-trained weights and keep it frozen during training.

**Analysis of Loss Functions.** Table 5 compares our proposed alignment loss with other auxiliary objectives used in SOTA methods. All experiments utilize the SeaFormer visual backbone. The standard segmentation loss ($\mathcal{L}_{\text{seg}}$) alone achieves 49.3 mIoU and 60.8 F-score. Adding the $\mathcal{L}_{\text{AVM}}$ (Zhou et al., 2022) yields inconsistent results: while F-score slightly improves, mIoU drops to 49.2, suggesting unstable optimization. More notably, the $\mathcal{L}_{\text{mix}}$ from AVSeg-Former (Gao et al., 2024) leads to performance degradation on both metrics (48.8 mIoU and 60.7 F-score), indicating that such complex constraints are ill-suited for lightweight networks with limited capacity. In contrast, our BCE-based $\mathcal{L}_{\text{msa}}$ consistently boosts performance (+1.1 mIoU and +1.8 F-score), confirming that direct, stable similarity supervision is the most effective strategy for lightweight AVS.

**Effectiveness of Key Components.** Table 6 validates the contribution of each module. The baseline (static concatenation) achieves 44.4 mIoU, which improves to 46.8 mIoU with Audio-Guided Visual Enhancement (AGVE). To validate our core Reciprocal Encoder, we compare two interaction strategies: simply adding the Decoder (CMFD) to AGVE yields 48.3 mIoU (static audio), whereas enabling Hierarchical Audio Refinement (HAR) to form the full Reciprocal Audio-Visual Encoder reaches 49.3 mIoU. This superiority (49.3 vs. 48.3) highlights that the dynamic, visually-refined audio state is more effective for semantic localization than static fusion. Finally, integrating the complete framework with CMFD and $\mathcal{L}_{\text{msa}}$ boosts performance to 50.4 mIoU. This confirms that while the encoder resolves

ambiguity, the decoder and loss are essential for grounding these semantics into precise boundaries.

**Comparison with Lightweight Interaction Alternatives.** Table 7 compares the proposed channel-wise broadcast with two sparse spatial interaction alternatives under the same setting. Although strided attention and top-$k$ attention reduce spatial computation, their performance is substantially lower than broadcast interaction. This indicates that our design is not merely a cheaper replacement for dense cross-attention; instead, the global audio state provides an effective semantic prior for lightweight AVS, while avoiding noisy token-level spatial matching.

**Robustness to Source Complexity.** Table 8 further analyzes MS3 samples according to the number of concurrent sounding sources. Both methods become less accurate as source complexity increases, but LightAVSeg remains consistently stronger across all groups. This supports that the compressed global audio state remains useful for union-level sounding-region localization, even though our model is not designed for fine-grained source-wise disentanglement.

**Ablation of Descriptor Pooling and Alignment Objective.** Table 9 studies two implementation choices in our alignment supervision. Max pooling outperforms average pooling, suggesting that it better preserves the most salient visible semantic evidence for audio refinement. For the alignment objective, BCE performs better than KL divergence, indicating that direct binary foreground supervision is more stable for lightweight segmentation networks.

**Transferability of Alignment Supervision.** Table 10 evaluates whether the alignment principle can transfer to other AVS models. Because the full multi-scale $\mathcal{L}_{\text{msa}}$ is tied to our hierarchical audio-state design, we implement an adapted single-scale variant for external R50-based models. The results show that the alignment principle can benefit

*Table 9.* Ablation of descriptor pooling and alignment objective on the MS3 benchmark.

| Group | Variant | $M_J \uparrow$ | $M_F \uparrow$ |
|---|---|---|---|
| Pooling | Average pooling | 49.2 | 61.2 |
| | Max pooling | **50.4** | **62.6** |
| Alignment loss | KL | 48.9 | 60.8 |
| | BCE | **50.4** | **62.6** |

*Table 10.* Transferability of adapted alignment supervision on the MS3 benchmark.

| Model | $M_J \uparrow$ | $M_F \uparrow$ |
|---|---|---|
| AVSBench-R50 + adapted $\mathcal{L}_{\mathrm{msa}}$ | 47.9 | 58.5 |
| AVSegFormer-R50 + adapted $\mathcal{L}_{\mathrm{msa}}$ | 49.8 | 62.5 |
| SelM-R50 + adapted $\mathcal{L}_{\mathrm{msa}}$ | **55.0** | **66.8** |

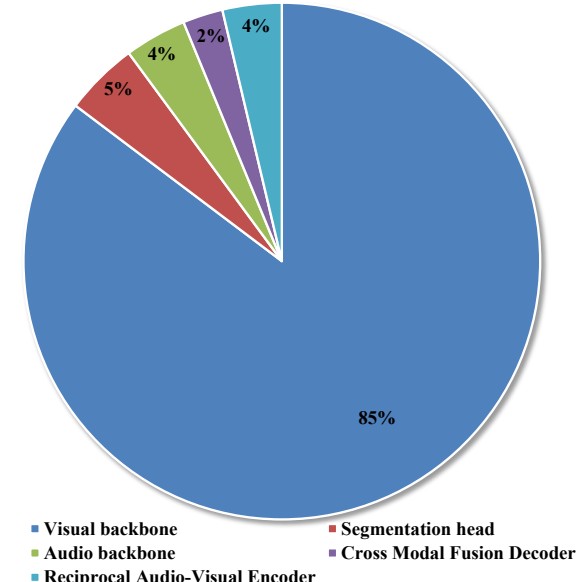

■ Visual backbone      ■ Segmentation head
■ Audio backbone      ■ Cross Modal Fusion Decoder
■ Reciprocal Audio-Visual Encoder

*Figure 4.* The inference latency of components.

architectures beyond LightAVSeg, while the original multi-scale formulation remains architecture-specific.

### 4.5. Qualitative Analysis

To provide visual insight into the feature learning process, Fig. 3 presents a comparison of feature activation maps from the last three stages across different methods on the MS3 benchmark. As observed in the first row, the AVS-Bench baseline is consistently distracted by background noise across all stages, resulting in scattered activations. AVSegFormer improves upon this by focusing better on the target object; however, its activations remain coarse and lack clear boundary definition. In contrast, our method demonstrates a distinct coarse-to-fine evolution across the network stages. Starting with broader context in earlier stages, our approach progressively suppresses background context in deeper stages to achieve precise alignment with the ground truth (GT). This visualization validates the effectiveness of our framework in accurately localizing sound sources with superior boundary precision compared to SOTA methods.

### 4.6. Latency Statistics

We analyze the component-wise latency on a Snapdragon 8 Elite as shown in Fig. 4. The visual backbone dominates the computation, consuming 85.3% (139.4 ms) of the total inference time. The audio backbone and segmentation head account for 3.9% (6.3 ms) and 4.6% (7.6 ms), respectively. Most importantly, our proposed interaction modules are extremely efficient: the Reciprocal Audio-Visual Encoder takes only 3.7% (6.1 ms), and the Cross-Modal Fusion Decoder (CMFD) occupies just 2.5% (4.1 ms). This validates that our lightweight design enables effective cross-modal fusion with negligible computational overhead.

## 5. Conclusion

In this work, we address the efficiency bottleneck in AVS by proposing LightAVSeg, a mobile-friendly framework tailored for edge deployment. Distinct from heavy attention-based architectures, we design a Reciprocal Audio-Visual Encoder that achieves linear interaction complexity through hierarchical global refinement. Complementing this, our Cross-Modal Fusion Decoder progressively injects these refined auditory cues back into the visual stream to guide precise segmentation. To further enhance performance without increasing inference cost, we introduce a training-only Multi-Scale Audio-Visual Alignment Loss that strictly enforces cross-modal semantic consistency. Experimental results on AVS benchmarks demonstrate that LightAVSeg establishes a new state-of-the-art for lightweight models. Crucially, we demonstrate efficient inference on a Snapdragon 8 Elite processor, proving the viability of high-performance AVS on resource-constrained devices.

## Acknowledgments

This work was primarily funded by the Independent Science and Technology Innovation Fund of Huazhong Agricultural University, Wuhan, Hubei, China (Grant No. 2662026XXQD001). Additional support was partially supported by the National Natural Science Foundation of China (Grant No. 62372150), the Hubei Key Research and Development Program of China (Grant No. 2024BAA008), and the Hubei Province Livestock Technology System (Grant No. 2025HBSTX4-13).

## Impact Statement

This paper presents work whose goal is to advance the field of Machine Learning. There are many potential societal consequences of our work, none which we feel must be specifically highlighted here.

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

## A. Limitations and Failure Cases

While LightAVSeg achieves state-of-the-art performance among lightweight models, it inevitably faces challenges due to the strict constraints on computational budget. As visualised in Fig. 5, we observe two primary types of failure cases:

1) Limited Fine-grained Detail. As shown in the second row (Incomplete Segmentation), the model sometimes fails to segment the entire object (e.g., missing parts of the horse). This is primarily attributed to the limited representational capacity of the lightweight visual backbone (SeaFormer). Unlike heavy backbones (e.g., ResNet-50) that maintain rich channel dimensions to encode complex texture details, our mobile-friendly encoder may discard subtle spatial cues during downsampling, leading to fragmented masks on large or textured objects.

2) Complex Multi-Source Ambiguity. In crowded scenes containing multiple instances of the same class (Top row) or occluded objects (Bottom row), the model may suffer from semantic inconsistency or missed detections. Since LightAVSeg compresses audio cues into a global state to achieve linear complexity, it may lack the dense audio-visual affinity matrices required to disentangle highly overlapped sounds or visually ambiguous targets. This reflects the inherent trade-off between the efficiency of global interaction and the precision of dense cross-modal attention.

Future Work. To mitigate these issues without increasing inference latency, future work could explore knowledge distillation techniques, transferring the fine-grained knowledge from a heavy teacher model (e.g., AVSegFormer) to our lightweight student network.

## B. Visualisation Results

Figure 6, Figure 7, and Figure 8 present qualitative results on the three major benchmarks: S4, MS3, and AVSS, respectively. Despite the presence of overlapping audio sources and complex visual backgrounds in these challenging sequences, our LightAVSeg exhibits great capability in accurately locating and delineating the specific sounding objects while suppressing silent ones. The visual examples clearly demonstrate the efficacy of our method in handling diverse and complex audio-visual scenarios.

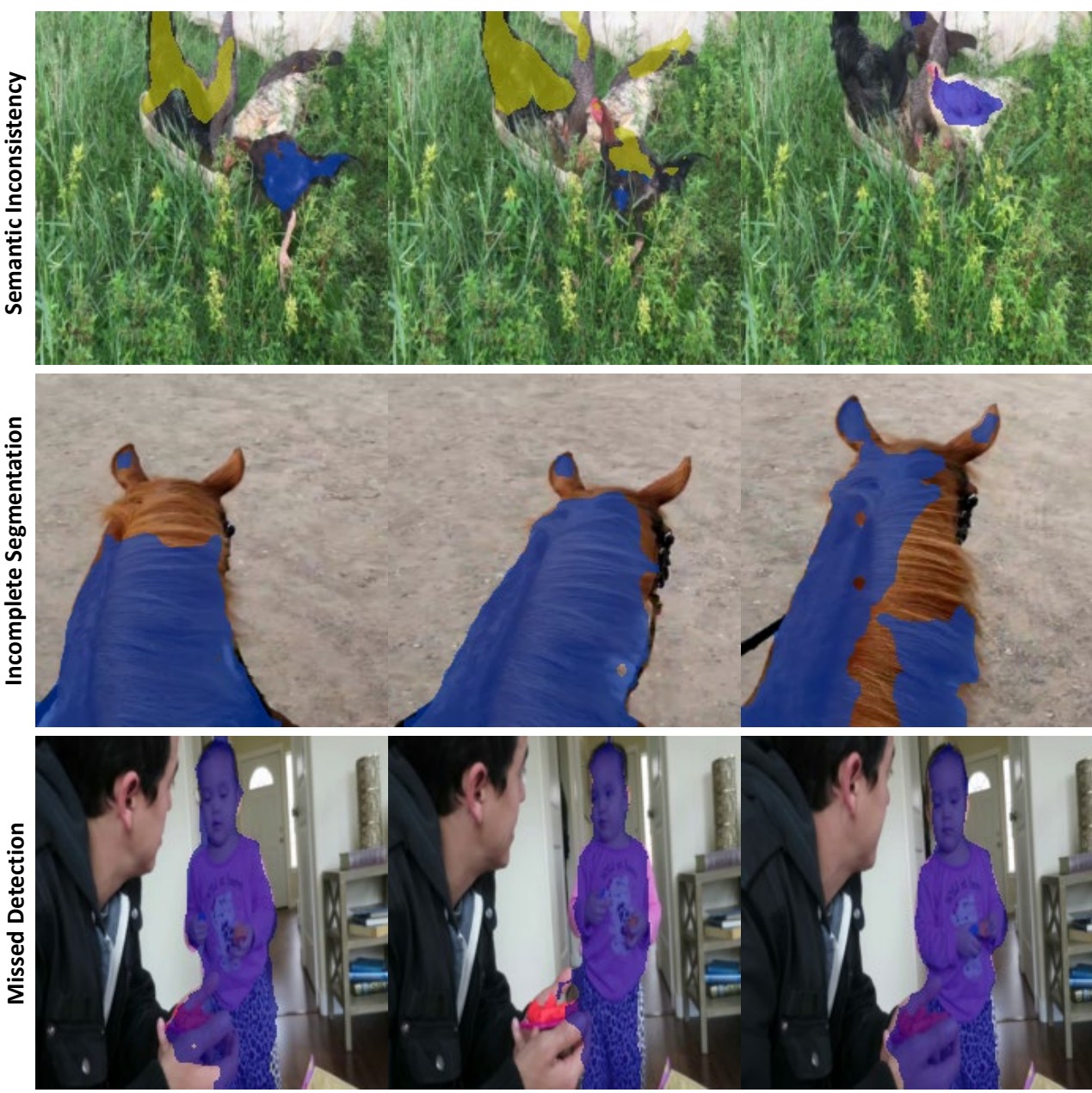

*Figure 5.* Failure cases of LightAVSeg. Due to the reduced capacity of the lightweight backbone, our model may struggle in challenging scenarios: (Top) Semantic inconsistency in crowded scenes with multiple similar objects; (Middle) Incomplete segmentation of large objects with complex textures; (Bottom) Missed detection when visual cues are subtle or occluded. These cases highlight the inherent trade-off between inference efficiency and fine-grained representational power.

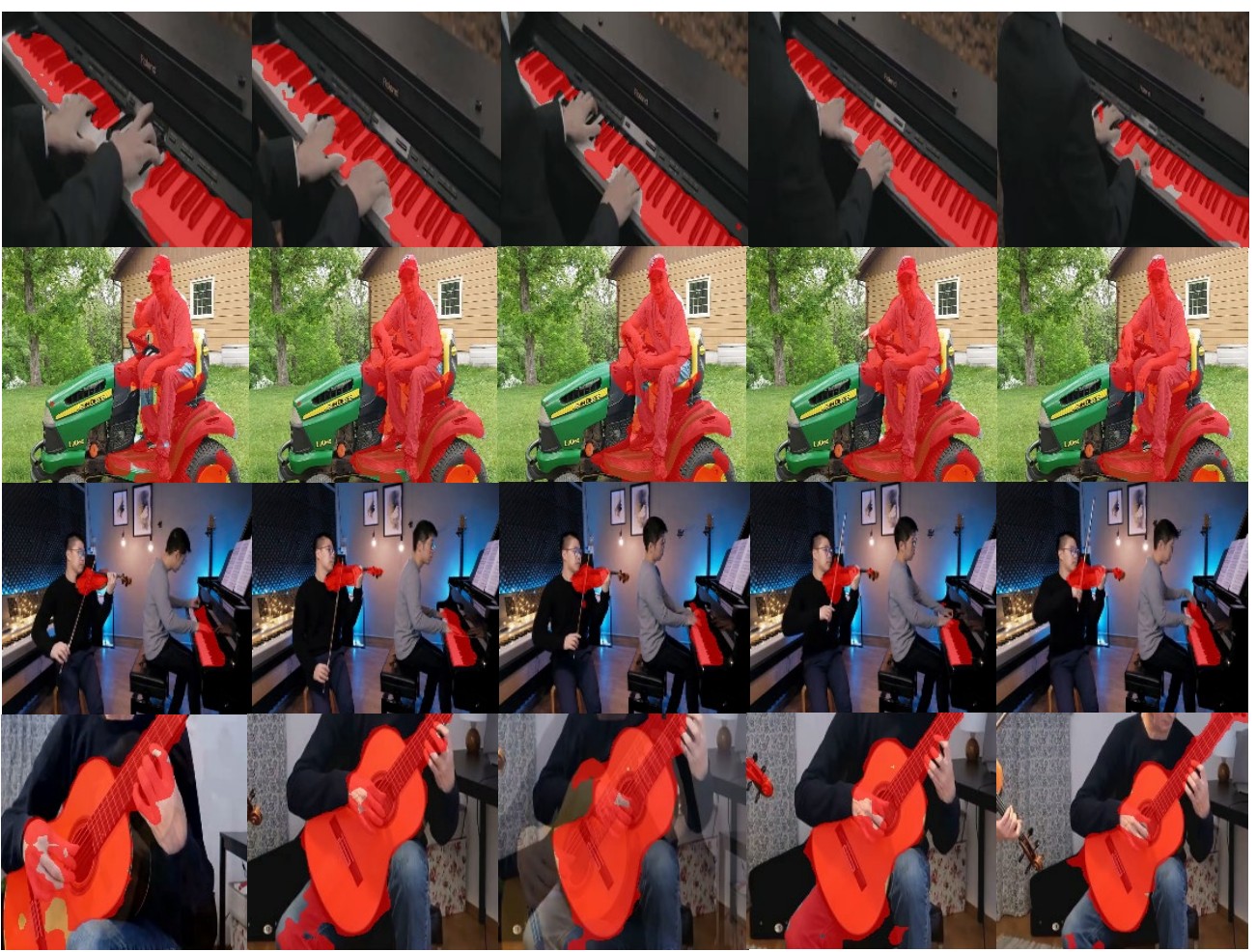

*Figure 6.* MS3 (Zhou et al., 2022) Visualisation Results.

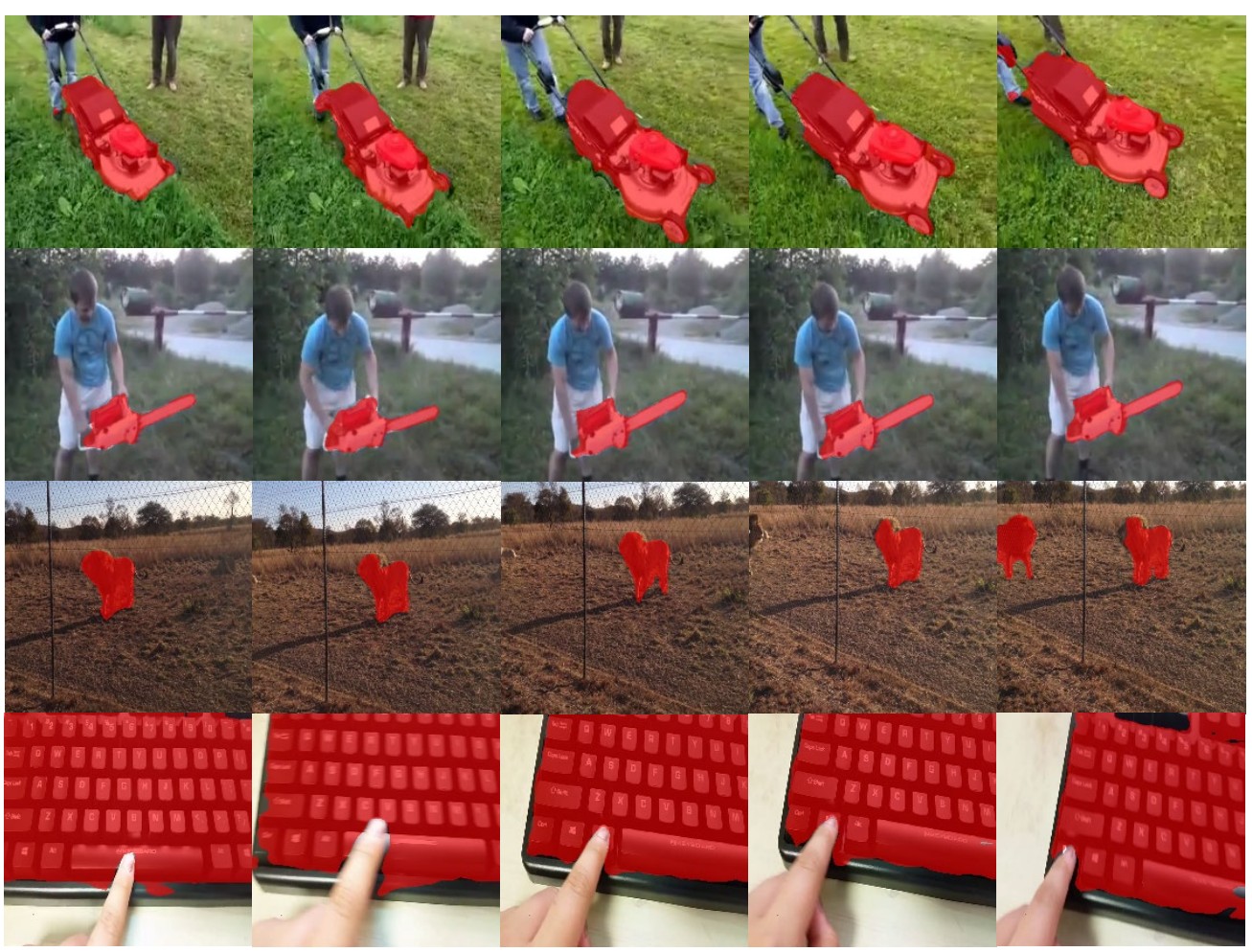

*Figure 7.* S4 (Zhou et al., 2022) Visualisation Results.

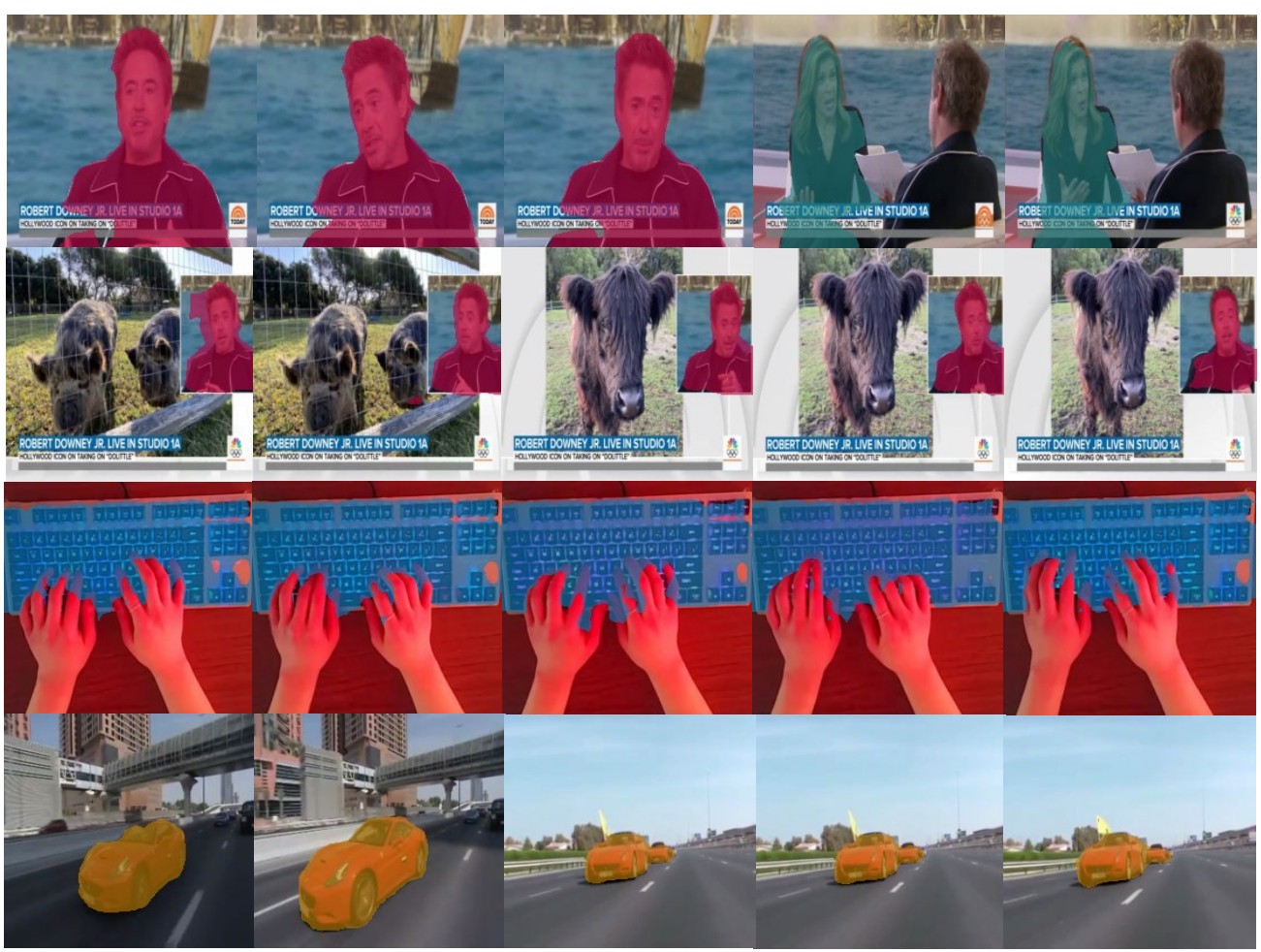

*Figure 8.* AVSS (Zhou et al., 2025a) Visualisation Results.

