# OpenReview forum: "LightAVSeg: Lightweight Audio-Visual Segmentation"
_ICML.cc/2026/Conference — ICML 2026 regular_

### Official Review · Reviewer_3Dfz · 2026-03-03

**Soundness:** 3
**Presentation:** 3
**Significance:** 2
**Originality:** 3
**Overall Recommendation:** 4
**Confidence:** 4

**Summary:**

This paper addresses the Audio-Visual Segmentation (AVS) problem, which aims to segment sounding objects at the pixel level. The authors argue that existing Transformer-based approaches are computationally heavy and therefore unsuitable for efficient deployment. To this end, they propose LightAVSeg, a lightweight AVS framework. The core components of LightAVSeg include a Reciprocal Audio-Visual Encoder and a Cross-Modal Fusion Decoder. The former performs hierarchical audio feature refinement through channel-wise modulation, while the latter injects audio guidance during cross-modal fusion. In addition, the paper introduces a Multi-Scale Audio-Visual Alignment loss to supervise the decoder. The proposed method is evaluated on three standard AVSBench subsets, and the results demonstrate its effectiveness.

**Compliance With Llm Reviewing Policy:**

Affirmed.

**Final Justification:**

I thank the authors for their rebuttal. The newly provided explanations with supportive experimental results well addressed my concerns. A lightweight audiovisual segmentation system is beneficial for real application. Overall, I decide to increase my final score to Weak Accept.

**Key Questions For Authors:**

Please refer to the weakness part

**Limitations:**

yes

**Strengths And Weaknesses:**

## Strengths
- The goal of developing an efficient audio-visual segmentation system is interesting and practically meaningful, particularly for real-world deployment scenarios. This aspect is often overlooked by mainstream methods.
- The proposed method achieves comparable performance in terms of mIoU and F-score while using significantly fewer parameters, demonstrating a good trade-off between accuracy and efficiency.
- The effectiveness of each main module and design choice is validated through ablation studies, making the experimental comparisons informative and convincing.

## Weaknesses
- The efficiency of the proposed method largely stems from its channel-wise design in the main modules. The manuscript repeatedly highlights the advantages of this channel-wise operation. For example, in Lines 211–212, the authors state that the method “suppresses audio channels that correspond to objects not visible in the current frame, ensuring $A_i$ becomes increasingly scene-specific.” A similar claim appears in Lines 188–191. However, these statements are not well supported by quantitative analysis, making it difficult to clearly understand or verify the actual benefits of the proposed design.
- Line 192 indicates that global visual features are obtained via max pooling. It would be helpful to include an ablation study comparing alternative strategies, such as average pooling. Additionally, in Lines 250–256, the authors briefly explain why KL divergence is not used to compute $\mathcal{L}_{msa}$. Providing corresponding experimental comparisons would further strengthen this justification.
- Most ablation experiments are conducted on the MS3 benchmark (Tables 3, 4, 5, and 6). Although MS3 is more challenging than S4 due to its multi-source setting, it contains a relatively small number of videos. Including more experimental results on the AVSS benchmark would make the evaluation more comprehensive and convincing.
- Figure 2 appears somewhat complex and difficult to follow. It would be helpful to simplify the visualization or provide clearer guidance in the main text to improve readability and better connect the figure with the methodological description.

---

> ### Author Rebuttal · Authors · 2026-03-30
>
> Thank you for the constructive comments on quantitative support, ablations, and presentation clarity. We address the four points below.
>
> **(1) Quantitative support for the channel-wise design.**
> To further substantiate the practical benefit of the channel-wise interaction, we compared our broadcast design with two representative lightweight spatial alternatives under the same setting:
>
> | Interaction design | MJ | MF |
> |:--|--:|--:|
> | Strided attention | 30.6 | 39.3 |
> | Top-k attention   | 44.7 | 57.2 |
> | Broadcast (ours)  | **50.4** | **62.6** |
>
> This shows that not all lightweight interaction schemes are equally effective, and that the proposed channel-wise broadcast is substantially stronger among the tested lightweight alternatives. In addition to the quantitative gap, the qualitative evidence is also clear: both the newly added response-map visualization(please refer to https://anonymous.4open.science/r/icml-rebuttal-0EDC/rebuttal%20(cropped)%20(pdfresizer.com).pdf) and the original Fig. 3 show that AVSegFormer and the sparse alternatives remain more diffuse/coarse and are more easily distracted by background regions, whereas our method exhibits a much clearer coarse-to-fine evolution, with activations progressively concentrating on the sounding regions and final masks that are noticeably sharper and closer to the ground truth.
>
> **(2) Max pooling vs. average pooling, and BCE vs. KL for `L_msa`.**
> We added the requested ablations for both the visual global descriptor construction and the alignment objective:
>
> | Pooling strategy | MJ | MF |
> |:--|--:|--:|
> | Average pooling | 49.2 | 61.2 |
> | Max pooling     | **50.4** | **62.6** |
>
> | Alignment loss | MJ | MF |
> |:--|--:|--:|
> | KL  | 48.9 | 60.8 |
> | BCE | **50.4** | **62.6** |
>
> These results support our design choices: max pooling better preserves the most salient visible semantic evidence for audio refinement, while BCE provides a more stable and effective supervision signal than KL in the lightweight setting.
>
> **(3) More evidence on AVSS.**
> To complement the MS3 ablations, we additionally report AVSS ablations for the key components:
>
> | AGVE | CMFD | HAR | `L_msa` | MJ | MF |
> |:--:|:--:|:--:|:--:|--:|--:|
> |  |  |  |  | 20.7 | 24.2 |
> | ✓ |  |  |  | 23.9 | 27.3 |
> | ✓ | ✓ |  |  | 26.6 | 31.5 |
> | ✓ |  | ✓ |  | 29.3 | 34.9 |
> | ✓ | ✓ | ✓ | ✓ | **30.6** | **36.4** |
>
> The same component-wise contribution trend therefore also holds on the larger AVSS benchmark, not only on MS3.
>
> **(4) Fig. 2 readability.**
> For presentation clarity, we will simplify Fig. 2 and add clearer stage-by-stage guidance in the main text so that the diagram aligns more directly with Eqs. (2)–(9).

---

> > ### Author Rebuttal · Reviewer_3Dfz · 2026-04-01
> >
> > I thank the authors for their rebuttal. The newly provided explanations with supportive experimental results well addressed my concerns. No more questions or suggestions from my side. I will increase my final score accordingly. Good luck.

---

> > > ### Author Response · Authors · 2026-04-03
> > >
> > > We sincerely thank the reviewer for the encouraging follow-up and for carefully considering our rebuttal. We are very grateful that the additional explanations and experiments addressed your concerns. We greatly appreciate your time, thoughtful feedback, and updated evaluation.

---

### Official Review · Reviewer_AmBL · 2026-03-08

**Soundness:** 2
**Presentation:** 2
**Significance:** 2
**Originality:** 2
**Overall Recommendation:** 2
**Confidence:** 4

**Summary:**

This paper proposes LightAVSeg, a lightweight framework for Audio-Visual Segmentation (AVS) targeting mobile deployment. The core idea is to replace the quadratic-complexity cross-modal attention used in existing AVS methods with a decoupled design: a Reciprocal Audio-Visual Encoder that performs semantic filtering via channel-wise gating (identifying what is sounding), and a Cross-Modal Fusion Decoder that handles spatial grounding (determining where to segment). Both modules operate at linear complexity O(N) by collapsing audio-visual interaction to global descriptors and channel-wise broadcasting, avoiding dense token-to-token affinity computation. An auxiliary Multi-Scale Audio-Visual Alignment Loss is introduced to enforce pixel-wise cross-modal consistency during training, which is discarded at inference. The method achieves 50.4 mIoU on MS3 with 20.5M parameters (~1/7 of AVSegFormer-R50) and 163.4 ms latency on a Snapdragon 8 Elite mobile CPU.

**Compliance With Llm Reviewing Policy:**

Affirmed.

**Key Questions For Authors:**

1. The Reciprocal Audio-Visual Encoder collapses all spatial information via max-pooling before audio-visual interaction. Can you provide evidence (e.g., from attention maps of existing dense-attention models) that spatial audio-visual affinity is indeed redundant? Have you tested a lightweight sparse attention alternative (e.g., top-k or strided attention) as a middle ground?

2. In the MS3 multi-source setting, how does the global audio state A_i disambiguate between multiple concurrent sound sources? Since A_0 is obtained by average pooling over the temporal window, it would blend multiple source signatures. Can you provide per-class or per-source-count breakdowns on MS3?

3. The lightweight baselines in Table 1 (AVSBench-Sea, AVSegFormer-Sea, SelM-Sea) appear to be your own re-implementations with swapped backbones. Were these baselines carefully tuned (e.g., learning rate, training schedule) to ensure a fair comparison, or were the original hyperparameters used directly? This matters because lightweight backbones often require different training recipes.

4. What is the total system FLOPs (not just backbone FLOPs)? Table 3 only reports backbone-level FLOPs. Since the paper's central claim is about interaction module efficiency, reporting the FLOPs of the interaction modules explicitly would be essential.

**Limitations:**

- The paper honestly acknowledges failure cases (Appendix A) related to limited backbone capacity and multi-source ambiguity, which is appreciated.
- However, several important limitations are not discussed:
  - The global audio pooling fundamentally limits the model's ability to handle fine-grained multi-source disentanglement, which is arguably the most important open challenge in AVS. This is an inherent architectural limitation, not just a capacity issue.
  - The evaluation is restricted to a single dataset family (AVSBench). Generalization to other audio-visual benchmarks or in-the-wild scenarios is unknown.

**Strengths And Weaknesses:**

**Soundness: 2 (Fair)**

Strengths:
- The formulation is generally correct. Equations (2)–(10) are mathematically consistent, and the complexity analysis (O(N) vs. O(N²)) is valid given the max-pooling to 1×1 before interaction.
- The claim that audio provides primarily global semantic cues rather than spatial information is a reasonable inductive bias for AVS, and the design follows logically from this observation.

Weaknesses:
- The theoretical justification for why channel-wise broadcasting (Eq. 3, Eq. 6) is sufficient to replace spatial cross-attention is thin. The paper asserts that "fine-grained localization boundaries are largely encoded in visual spatial features" but provides no formal or empirical analysis (e.g., mutual information, attention map statistics from existing models) to validate this assumption. If the sounding object occupies a small region and shares channel statistics with background objects, a global channel bias cannot disambiguate them spatially — yet this failure mode is not analyzed.
- The Reciprocal Audio-Visual Encoder (Eq. 2) is essentially a SE-block-style gating mechanism applied between modalities. The novelty of this operation over standard squeeze-and-excitation or FiLM conditioning is not clearly articulated.
- The L_msa loss (Eq. 8–9) computes cosine similarity between a global audio vector and per-pixel visual features, then supervises with the foreground mask. This implicitly assumes that all foreground pixels should have high cosine similarity with the audio embedding — a strong assumption that breaks down when multiple visually distinct objects produce sound simultaneously (the MS3 setting). The paper does not discuss this limitation.
- The comparison in Table 1 is somewhat unfair: the "lightweight" baselines (AVSBench-Sea, AVSegFormer-Sea, SelM-Sea) are created by the authors by simply swapping backbones, not official lightweight variants. The actual contribution of the proposed interaction module vs. better engineering of the backbone swap is unclear.

**Presentation: 2 (Fair)**

Strengths:
- The paper is generally well-organized with a clear pipeline figure (Fig. 2) and the efficiency-accuracy trade-off plot (Fig. 1) is informative.
- The qualitative comparison in Fig. 3 effectively illustrates the progressive refinement behavior.

Weaknesses:
- Key details are missing: how is temporal modeling handled? The paper mentions T frames but all equations operate on single-frame features. The temporal dimension seems to be treated as a batch dimension, which should be stated explicitly.
- The related work section has citation issues: SegFormer, PiCo, and "prototype-based segmentation" are all cited as (Xie et al., 2021), which appears to be a single paper (SegFormer). This is either a citation error or conflation of distinct works.
- The paper lacks a FLOPs comparison at the system level (only backbone FLOPs in Table 3). Total FLOPs including the interaction modules would strengthen the efficiency argument.

**Significance: 2 (Fair)**

- The practical contribution — enabling AVS on mobile devices — is relevant and timely. However, the performance gains over the naive backbone-swap baselines, while substantial, largely come from better architectural integration rather than a fundamentally new insight. The core techniques (channel-wise gating, global pooling, deep supervision) are well-established.
- The MS3 result (50.4 mIoU) only marginally exceeds AVSegFormer-R50 (49.5), and on S4 it is slightly below (75.6 vs. 76.5). The efficiency gains are clear, but the accuracy improvements are modest.

**Originality: 2 (Fair)**

- The individual components — SE-style gating for cross-modal fusion, channel-wise broadcasting, multi-scale BCE alignment loss, deep supervision — are all known techniques. The contribution is primarily in their combination and application to the lightweight AVS setting.
- The "decoupled what/where" framing is conceptually clean but not technically novel; similar decompositions exist in visual grounding and referring segmentation literature.
- The auxiliary alignment loss is the most novel element, but its formulation (cosine similarity + BCE) is straightforward.

---

> ### Author Rebuttal · Authors · 2026-03-30
>
> Thank you for the detailed and critical feedback. We address the main concerns below.
>
> **(Soundness W1, Q1) Why channel-wise broadcasting is a reasonable lightweight choice.**
>
> We follow the reviewer’s suggestion and implement two representative sparse spatial alternatives:
>
> | Interaction design | MJ | MF |
> |:--|--:|--:|
> | Strided attention | 30.6 | 39.3 |
> | Top-k attention | 44.7 | 57.2 |
> | Broadcast (ours) | **50.4** | **62.6** |
>
> The results show that not all lightweight interaction schemes are equally effective. The qualitative evidence is also consistent: the alternative response map visualizations (https://anonymous.4open.science/r/icml-rebuttal-0EDC/rebuttal%20(cropped)%20(pdfresizer.com).pdf) as well as our original Fig. 3 show that AVSegFormer and the sparse alternatives remain more diffuse/coarse and are more easily distracted by background regions, whereas our method exhibits a clearer coarse-to-fine evolution and sharper final localization.
>
> **(Soundness W2) Why the contribution is not merely an SE/FiLM-style gate.**
>
> Our contribution is **not** introduction of a new gating primitive, but an **architecture-level design** in which a global audio state is **recursively updated across stages** using stage-specific visual semantics, and then reused as a semantic prior throughout the hierarchy and decoder.
>
> In contrast, SE/FiLM-style mechanisms are typically **one-shot, per-layer modulations** of the current feature map. The ablation supports this distinction: static fusion (**AGVE + CMFD**) yields **48.3** mIoU, while enabling dynamic **HAR** further improves this to **49.3**.
>
> **(Soundness W3, Q2, L1) Multi-source ambiguity and the limitation of a global audio cue.**
>
> Our design is not centered around **source-wise disentanglement**. On MS3, the supervision target is a **binary sounding-region mask**, so the role of the global audio state is to provide a **union-level semantic cue** indicating which regions are acoustically relevant, while the visual stream/decoder performs the actual spatial grounding and boundary refinement. To directly examine this issue, we additionally analyzed MS3 by source complexity:
>
> | Method | 1 source | 2 sources | >3 sources |
> |:--|--:|--:|--:|
> | AVSegFormer | 44 / 53 | 34 / 45 | 32 / 49 |
> | Ours | **53 / 64** | **45 / 58** | **44 / 65** |
>
> Thus, both methods degrade with source complexity, but ours remains consistently stronger. We will state this explicitly as a limitation: our method is effective for **union-level localization**, but not designed for fine-grained source-wise disentanglement.
>
> **(Soundness W4, Q3) Fairness of the lightweight baselines.**
> The lightweight baselines in Table 1 are **controlled lightweight re-implementations** under the same SeaFormer + MobileNetV2 regime, rather than official released lightweight variants. To reduce the concern that our gain mainly comes from under-tuned backbone swaps, we performed a consistent tuning sweep on the strongest lightweight baselines and our model. The best tuned results on MS3 are **41.0/51.2** for AVSegFormer-Sea, **45.7/57.6** for SelM-Sea, and **50.6/63.2** for ours (MJ/MF), compared with the originally reported **40.7/50.7**, **45.7/57.6**, and **50.4/62.6**, respectively. The relative ranking therefore remains unchanged after tuning. (We provide the full tuning table in our response to **Reviewer WjPX(1)**.)
>
> **(Presentation W3, Q4) System-level FLOPs and reporting clarity.**
> We additionally profiled the full model and obtained **2.70G total FLOPs**, with the following breakdown:
>
> | Component | FLOPs | Ratio |
> |:--|--:|--:|
> | Audio branch | 0.0718G | 2.66% |
> | Visual backbone | 2.371G | 87.81% |
> | Interaction modules | 0.2562G | 9.49% |
> | Segmentation head | 0.0012G | 0.04% |
>
> This directly supports our claim that the interaction modules are lightweight at the **system level**, not only relative to the backbone.
>
> **(Presentation W1, W2) Presentation details.**
> The visual branch does **not** use an explicit cross-frame temporal module; in implementation, the synchronized frame dimension is folded into the batch dimension for efficient parallel processing. We will also correct the related-work citation issues.
>
> **(L2) Evaluation scope and generalization.**
> AVS benchmark is the standard benchmark for pixel-level AVS, and our paper explicitly defines its scope within this established task. We will leave out-of-domain or **in-the-wild** generalization as future work.

---

### Official Review · Reviewer_1154 · 2026-03-10

**Soundness:** 3
**Presentation:** 2
**Significance:** 3
**Originality:** 2
**Overall Recommendation:** 4
**Confidence:** 2

**Summary:**

This paper proposes LightAVSeg, a lightweight AVS framework designed for the mobile end. To address the efficiency bottleneck in cross-modal interaction modules of existing methods, the paper decouples interaction into semantic filtering at the encoding stage and spatial grounding at the decoding stage, along with an auxiliary alignment loss during training to mitigate semantic ambiguity in lightweight networks. The proposed framework achieves lightweight SOTA on all AVSBench benchmarks and less inference latency on Snapdragon 8 Elite.

**Compliance With Llm Reviewing Policy:**

Affirmed.

**Final Justification:**

The authors' response has addressed my questions.

**Key Questions For Authors:**

See the weakness section.

**Limitations:**

The authors noted that there are many potential social consequences, but none are essential to the discussion.

**Strengths And Weaknesses:**

### Strengths
(1)  The paper accurately identifies that the efficiency bottleneck in AVS lies in the interaction module rather than the backbone. Proposed module design is well-aligned with the motivation.

(2) Strong results with better performance at lower cost, demonstrating a clear efficiency-accuracy advantage.

### Weaknesses
(1) Missing efficiency-performance comparison across different spatial resolutions N. I may have missed it, but seems that all experiments are conducted at a single 224×224 resolution, with no reported latency or accuracy comparison under different resolution.
(2) Ablations demonstrate effectiveness, but do not sufficiently verify efficiency or generalizability. The ablation studies clearly demonstrate the effectiveness of each module, and the method does bring performance improvements.

However, I have two concerns:

(a) The current experiments show that each module is effective (i.e., better with than without), but this alone does not seem sufficient to demonstrate that these specific design choices are efficient or optimal compared to other lightweight alternatives.

(b) The auxiliary loss L_msa is compared against other losses within the proposed framework, and the results are clear. However, I am curious whether L_msa is equally effective on other AVS models. Validating its effect on at least one other method would better support its generalizability as an independent contribution.

---

> ### Author Rebuttal · Authors · 2026-03-30
>
> Thank you for the constructive feedback and for recognizing the core motivation of the paper. We address the three concerns below.
>
> **(1) Resolution-dependent accuracy-efficiency trade-off.**
> We additionally evaluated AVSegFormer, SelM, and our method at **224x224, 512x512, and 1024x1024** input resolutions on MS3, reporting both **accuracy** and **latency**. All methods improve as resolution increases, but at a substantially higher inference cost:
>
> | Method | Resolution | MJ | MF | Latency (ms) |
> |:--|:--:|--:|--:|--:|
> | AVSegFormer | 224  | 40.7 | 50.7 | 432.6 |
> | AVSegFormer | 512  | 45.2 | 55.4 | 2422.6 |
> | AVSegFormer | 1024 | 50.1 | 60.8 | 8738.5 |
> | SelM        | 224  | 45.7 | 57.6 | 308.6 |
> | SelM        | 512  | 51.8 | 63.1 | 1666.4 |
> | SelM        | 1024 | 57.6 | 68.3 | 6820.1 |
> | Ours        | 224  | 50.4 | 62.6 | **163.4** |
> | Ours        | 512  | 56.2 | 67.8 | 896.8 |
> | Ours        | 1024 | **59.0** | **71.5** | 3561.3 |
>
> Our gains are not tied to a single 224×224 operating point.  Our method consistently yields the strongest accuracy while also maintaining a clear latency advantage across all three resolutions.
>
> **(2) Effectiveness vs. design choice.**
> Standard with/without ablations show that a module is useful, but they do not by themselves establish whether this specific lightweight design is preferable to other lightweight alternatives. To address this, we implemented two representative sparse spatial alternatives under the same setting: **strided attention** and **top-k attention**:
>
> | Interaction design | MJ | MF |
> |:--|--:|--:|
> | Strided attention | 30.6 | 39.3 |
> | Top-k attention   | 44.7 | 57.2 |
> | Broadcast (ours)  | **50.4** | **62.6** |
>
> This shows that not all lightweight interaction schemes are equally effective, and that the proposed channel-wise broadcast is substantially stronger among the tested alternatives. This is also consistent with the qualitative evidence: both the newly added response-map visualization (please refer to https://anonymous.4open.science/r/icml-rebuttal-0EDC/rebuttal%20(cropped)%20(pdfresizer.com).pdf) and the original Fig. 3 show that AVSegFormer and the sparse alternatives remain more diffuse/coarse and are more easily distracted by background regions, whereas our method exhibits a much clearer coarse-to-fine evolution and sharper final localization.
>
> **(3) Generalizability of `L_msa`.**
> The original multi-scale `L_msa` is tied to our hierarchical audio-state design and supervises multiple decoder scales, so it cannot be directly transplanted to models without multi-scale audio states. To test whether the **underlying alignment principle** transfers beyond our architecture, we implemented an **adapted single-scale variant** on external AVS models:
>
> | Model | MJ | MF |
> |:--|--:|--:|
> | AVSBench-R50 + adapted `L_msa`    | 47.9 | 58.5 |
> | AVSegFormer-R50 + adapted `L_msa` | 49.8 | 62.5 |
> | SelM-R50 + adapted `L_msa`        | **55.0** | **66.8** |
>
> These results suggest that the alignment principle is transferable beyond our model, while the original full multi-scale `L_msa` remains architecture-specific. We will revise the paper to make this distinction explicit rather than overstating plug-and-play generality.

---

> > ### Author Rebuttal · Reviewer_1154 · 2026-04-04
> >
> > The authors' response has addressed my questions.

---

> > > ### Author Response · Authors · 2026-04-04
> > >
> > > We sincerely thank you for taking the time to carefully read our rebuttal. If there are any aspects of our response that require further clarification or if you have any additional questions, we are more than happy to engage in further discussion.

---

### Official Review · Reviewer_WjPX · 2026-03-10

**Soundness:** 3
**Presentation:** 3
**Significance:** 3
**Originality:** 3
**Overall Recommendation:** 5
**Confidence:** 3

**Summary:**

The paper introduces LightAVSeg, a lightweight audio-visual segmentation model designed to run efficiently on edge devices like mobile phones. Current top models use dense cross-modal attention mechanisms that are computationally expensive. The authors get around this by splitting the task: they use a Reciprocal Audio-Visual Encoder to figure out what is making the sound using global features, and a Cross-Modal Fusion Decoder to figure out where it is spatially. This drops the interaction cost from quadratic to linear. They also include an auxiliary alignment loss during training to aid consistency, which doesn't impact inference time.

**Compliance With Llm Reviewing Policy:**

Affirmed.

**Final Justification:**

The authors have addressed all my concerns and I will maintain my positive score. I look forward to seeing the final manuscript with all the improvements made during the discussion period.

**Key Questions For Authors:**

In Figure 3, the baseline activation maps for AVSBench and AVSegFormer look incredibly noisy and distracted by the background. Were these baselines fully tuned and optimized for this specific setup, or were they just run out-of-the-box? If you can confirm the baselines were fairly optimized, it would reassure that your massive mIoU gains are genuinely due to your architecture and not just a weak baseline comparison.

You mentioned in the limitations section that compressing audio into a global state makes it hard to disentangle overlapping sounds or visually ambiguous targets. Since the MS3 benchmark is specifically designed around complex scenes with concurrent sound sources , how exactly is the model still managing to hit 50.4 mIoU on it? A brief explanation of why the global compression doesn't completely fail on MS3 would really strengthen the contribution of the Reciprocal Audio-Visual Encoder.

You showed that your BCE-based auxiliary alignment loss ($\mathcal{L}_{msa}$) works much better for lightweight networks than the complex losses used by heavier models like AVSegFormer. Have you tried plugging your $\mathcal{L}_{msa}$ loss into those heavier baselines to see if it fixes their messy boundary definitions? If it improves the heavy models too, that would make your loss function a much more significant contribution to the broader field.

**Limitations:**

Yes the limitations were discussed. However, given that this is an audio-visual localization and segmentation model designed for deployment on mobile devices, acknowledging potential privacy or surveillance implications, even briefly, would help make the impact statement more complete.

**Strengths And Weaknesses:**

Soundness

The architectural shift from $\mathcal{O}(N^{2})$ cross-modal attention to an $\mathcal{O}(N)$ global channel-wise modulation is mathematically sound and empirically validated. The ablation studies clearly justify their design choices, such as picking SeaFormer-Large and MobileNetV2 to balance accuracy and mobile latency. The global audio compression method fundamentally limits the model's ability to disentangle highly overlapped sounds or similar sources in crowded scenes. This is acknowledged in their failure cases. Furthermore, the lightweight visual backbone drops fine-grained details on large or textured objects.

Presentation

The paper is straightforward and the narrative flows logically. Figure 2 maps out the dual-stream architecture cleanly , and the component-wise latency breakdown in Figure 4 is extremely helpful. Figure 3 also clearly illustrates how the model progressively filters out the background noise that distracts the baselines.

Significance

Enabling audio-visual segmentation on edge devices is a highly practical problem with real-world applications. Achieving a latency of 163.4 ms on a mobile CPU while reaching 50.4 mIoU on the complex MS3 benchmark proves this approach has serious utility.

Originality

Decoupling the interaction into semantic filtering and spatial grounding to avoid dense spatial audio maps is a clever, pragmatic approach. Additionally, introducing the auxiliary Multi-Scale Audio-Visual Alignment Loss as a structural prior that explicitly aligns audio with visual regions proves to be a highly effective training strategy for lightweight models.

---

> ### Author Rebuttal · Authors · 2026-03-30
>
> Thank you for the positive assessment and the very helpful questions. We address the three points most central to this review below.
>
> **(1) Fairness of the Figure 3 baseline comparison.**
> The lightweight baselines in Table 1 / Fig. 3 are **controlled lightweight re-implementations** under the same SeaFormer + MobileNetV2 regime, rather than official released “light” variants. To verify that the qualitative gap is not caused by weak tuning, we performed a consistent sweep on the strongest lightweight baselines and our model, varying the learning rate in {2e-5, 1e-4, 1e-3} and the training length in {50, 150, 250} epochs under the same validation protocol:
>
> | Method | LR | Epochs | MJ | MF |
> |:--|:--:|:--:|--:|--:|
> | AVSegFormer-Sea | 2e-5 | 50  | 40.2 | 49.2 |
> | AVSegFormer-Sea | 1e-4 | 50  | 40.7 | 50.7 |
> | AVSegFormer-Sea | 1e-3 | 50  | 39.1 | 45.4 |
> | AVSegFormer-Sea | 1e-4 | 150 | **41.0** | **51.2** |
> | AVSegFormer-Sea | 1e-4 | 250 | **41.0** | **51.2** |
> | SelM-Sea        | 2e-5 | 50  | 45.3 | 56.9 |
> | SelM-Sea        | 1e-4 | 50  | **45.7** | **57.6** |
> | SelM-Sea        | 1e-3 | 50  | 42.6 | 53.4 |
> | SelM-Sea        | 1e-4 | 150 | **45.7** | **57.6** |
> | SelM-Sea        | 1e-4 | 250 | **45.7** | **57.6** |
> | Ours            | 2e-5 | 50  | 49.6 | 60.5 |
> | Ours            | 1e-4 | 50  | 50.4 | 62.6 |
> | Ours            | 1e-3 | 50  | 47.2 | 59.5 |
> | Ours            | 1e-4 | 150 | 50.4 | 62.6 |
> | Ours            | 1e-4 | 250 | **50.6** | **63.2** |
>
> The relative ranking remains unchanged after tuning, so our gain is not explained by an under-optimized baseline. We will revise the manuscript to state explicitly that these are **controlled lightweight comparisons**.
>
> **(2) Why global audio compression does not fail on MS3.**
> Our model is not designed for **source-wise disentanglement**. On MS3, the supervision target is a **binary sounding-region mask**, so the role of the global audio state is to provide a **union-level semantic cue** indicating which regions are acoustically relevant, while the visual hierarchy/decoder performs the actual spatial grounding and boundary refinement. To directly test this issue, we additionally analyzed MS3 by source complexity:
>
> | Method | 1 source | 2 sources | >3 sources |
> |:--|--:|--:|--:|
> | AVSegFormer | 44 / 53 | 34 / 45 | 32 / 49 |
> | Ours        | **53 / 64** | **45 / 58** | **44 / 65** |
>
> Thus, both methods degrade as the scene becomes more complex, but ours does not collapse under concurrent sources and remains consistently stronger across all source-count regimes. We will make this scope explicit in the revision: our method is effective for **union-level sounding-object localization**, but not designed for fine-grained source-wise disentanglement.
>
> **(3) Transferability of `L_msa`.**
> The original multi-scale `L_msa` is tied to our hierarchical audio-state design, so it is not a plug-and-play module that can be inserted unchanged into arbitrary AVS architectures. To test whether the **underlying alignment principle** transfers beyond our model, we implemented an **adapted single-scale variant** on external R50 baselines:
>
> | Model | MJ | MF |
> |:--|--:|--:|
> | AVSBench-R50 + adapted `L_msa`    | 47.9 | 58.5 |
> | AVSegFormer-R50 + adapted `L_msa` | 49.8 | 62.5 |
> | SelM-R50 + adapted `L_msa`        | **55.0** | **66.8** |
>
> These results indicate that the alignment idea itself is useful beyond our architecture, while the original full multi-scale formulation remains architecture-specific. We will revise the wording accordingly to avoid overstating plug-and-play generality.
>
> Finally, thank you for the suggestion on broader impact. We will expand the impact statement to explicitly mention potential **privacy / surveillance risks** and the need for responsible deployment of edge AVS systems.

---

> > ### Author Rebuttal · Reviewer_WjPX · 2026-03-31
> >
> > Thank you for the comprehensive and data-driven rebuttal. You directly and effectively addressed my questions.
> >
> > The hyperparameter sweeps confirm that the performance gains are architectural rather than due to under-optimized baselines. I also appreciate the clear distinction you drew between union-level localization and source-wise disentanglement for the MS3 results; making this limitation explicit in the revision is the right call. Additionally, showing that the alignment principle of the $\mathcal{L}_{msa}$ loss transfers to heavier baselines adds solid value to the paper.
> >
> > My concerns are fully resolved, and I appreciate the diligence you put into addressing this feedback.

---

> > > ### Author Response · Authors · 2026-04-03
> > >
> > > We sincerely thank the reviewer for the positive follow-up and for carefully considering our rebuttal. We are very grateful that our additional experiments and clarifications addressed your concerns. We greatly appreciate your time and thoughtful evaluation.

---

### Decision · Program_Chairs · 2026-04-30

**Decision:**

Accept (regular)

**Comment:**

Focusing on the reviews that engaged with the authors’ rebuttal, the assessments are now consistently positive, with one Accept and two Weak Accepts. The main concern previously raised was about limitations and clarity of the approach, but the rebuttal helped clarify the scope and addressed the core issues. Other reviewers found the method technically sound and the experimental evidence convincing. Overall, I recommend Accept.